# Quantification and Optimization of Ethanolic Extract Containing the Bioactive Flavonoids from *Millettia pulchra* Radix

**DOI:** 10.3390/molecules26123641

**Published:** 2021-06-15

**Authors:** Thanh-Hoa Vo, Chia-Ching Liaw, Yu-Chi Lin, Duc Hanh Nguyen, Thi Tuyet Nhung Nguyen, Ching-Kuo Lee, Yao-Haur Kuo

**Affiliations:** 1Ph.D. Program in Clinical Drug Development of Herbal Medicine, College of Pharmacy, Taipei Medical University, Taipei 11031, Taiwan; d339107005@tmu.edu.tw (T.-H.V.); d339109002@tmu.edu.tw (T.T.N.N.); 2National Research Institute of Chinese Medicine, Taipei 11221, Taiwan; liawcc@nricm.edu.tw; 3School of Medicine, Vietnam National University Ho Chi Minh City, Ho Chi Minh City 700000, Vietnam; 4Department of Biochemical Science and Technology, National Chiayi University, Chiayi 60004, Taiwan; 5Department of Marine Biotechnology and Resources, National Sun Yat-sen University, Kaohsiung 80424, Taiwan; m8952612@hotmail.com; 6Department of Pharmaceutics, Faculty of Pharmacy, University of Medicine and Pharmacy at Ho Chi Minh City, Ho Chi Minh City 700000, Vietnam; duchanh@ump.edu.vn; 7Faculty of Pharmacy, Lac Hong University, Dong Nai 810000, Vietnam; 8School of Pharmacy, College of Pharmacy, Taipei Medical University, Taipei 11031, Taiwan; 9Graduate Institute of Integrated Medicine, College of Chinese Medicine, China Medical University, Taichung 40402, Taiwan

**Keywords:** flavonoids, quantification, optimization, response surface methodology, artificial neural network

## Abstract

*Millettia pulchra* is traditionally used for treating diseases, including joint pain, fever, anemia, and allergies. It is also a potential resource of natural flavonoid derivatives, which represents major constituents of this plant. This study aimed to isolate the major compounds from *M. pulchra* radix, develop and validate the HPLC-PDA method to determine their contents, and optimize its extraction. Four major flavonoid derivatives (karanjin, lanceolatin B, 2”,2”-dimethylpyrano-[5″,6″:7,8]-flavone, and pongamol) were isolated using silica gel column chromatography, crystallization techniques in large amounts with high purities (>95%). A simple, accurate high-performance liquid chromatography–photodiode array (HPLC–PDA) detection method has been developed and validated with significantly statistical impacts according to International Conference on Harmonization (ICH) guidelines. The Response Surface Methodology (RSM), Artificial Neural Network (ANN) models were employed to predictive performance and optimization of the extraction process. The optimized conditions for the extraction of major flavonoids were: extraction time (twice), solvent/material ratio (9.5), and ethanol concentration (72.5%). Our research suggests an effective method, which will be helpful for quality control in the pharmaceutical development of this species.

## 1. Introduction

*Millettia pulchra* Kurz (Fabaceae) radix has been traditionally used to treat rheumatic arthritis, infantile malnutrition, dizziness, anemia, and allergies [1,2,3]. Recent reports have revealed that its radix also has the pharmacological effects such as cardiovascular protection, hepatoprotection and anti-tumor properties [1,2]. These bio-activities have been associated with a number of isolated flavonoids, such as karanjin, lanceolatin B and pongamol [4,5,6].

The unique furanoflavonoid karanjin was first isolated from *Pongamia pinnata* (L.) Pierre [6]. The reports on in vitro and in vivo studies of karanjin suggests that it exhibits significant biological effects, including anti-tumor, anti-inflammatory, anti-diabetic, anti-oxidant, and anti-colitis activities [1,2,6,7]. These medicinal properties are mainly attributed to its anti-oxidant effect via the TNF-α signaling pathway. Karanjin has been also demonstrated to stimulate GLUT4 translocation to plasma via triggering AMP-activated protein kinase (AMPK), without including PI 3-kinase and protein kinase B (AKT), which modulates TNF and insulin signaling, plays a vital role in cell cycle arrest, metabolism, and drug efflux, and inhibits ROS [4,6].

Lanceolatin B, one of the major components of *M. pulchra*, shows anti-inflammatory activity in vitro via the inhibition of IL-1β, IL-6, TNF-α, COX-2, iNOS, NF-κB, and p-IκKα [5]. It has also been proved to block the cell cycle of CYP1A1 (Cytochrome P450 Family 1 Subfamily A Member 1)-overexpressing MCF-7 breast cancer cells [7]. Thus, lanceolatin B may have potential as a cancer chemopreventive and anti-inflammatory agent.

There is a lack of studies on bio-activity of 2″,2″-dimethylpyrano-[5″,6″:7,8]-flavone. This pyranoflavone could reduce the nitric accumulation in LPS-stimulated RAW264.7 cells [5].

Pongamol [(2*Z*)-3-hydroxy-1-(4-methoxy-1-benzofuran-5-yl)-3-phenylprop-2-en-1-one] was also first isolated from the seed oil of *P. pinnata* [4]. Later, it was also isolated from several species of the Fabaceae family [1,2,8]. A number of studies describing its anti-hyperglycemic [9], anti-myocardial ischemia [10,11] and anti-oxidant effects [12] of pongamol have been reported. Furthermore, it possesses significant antiproliferative and cancer chemopreventive activity [7]. According to the above bio-evidence survey, karanjin, lanceolatin B, pongamol, and 2″,2″-dimethylpyrano-[5″,6″:7,8]-flavone may serve as bio-markers for *M. pulchra* radix.

The natural product field has widely applied Response Surface Methodology (RSM) to optimize bioactive constituent’s extraction. In RSM, the relative importance of each independent variable is evaluated, and optimal operating conditions are determined for each response. The aim of RSM is to reduce the number of experiments required and to determine how different factors affect the experiments [13,14]. Recently, there are several reports on Artificial Neural Network (ANN) as a powerful modelling tool to estimate and optimize the extraction process [15,16]. The great advantage of ANNs is that it is efficient to solve nonlinear multivariate regression problems evenly when the relationship’s extract nature is unknown [16]. ANN modeling is suitable for studying the relationship between independent variables and responses of the extraction process using a limited number of experimental runs [15]. A comparison of RSM and ANN models could be a helpful approach to achieve higher accuracy and efficiency on optimization of biochemical extraction [16,17].

Consequently, our work aims to isolate and quantify the major bioactive flavonoids from *M. pulchra* radix and optimize the extraction process to obtain the optimal condition for extracting the mentioned isolates. Besides that, observing the variation of major components’ content in *M. pulchra* radix according to the age of the plant helps to determine the time for harvesting this medicinal plant.

## 2. Results and Discussion

### 2.1. Extraction and Isolation of Major Compounds

Four major compounds (Figure 1) were successfully isolated from the radix of *M. pulchra* and purified via the simple isolation method. By comparison with the literature data, the structures of these isolated compounds were elucidated as lanceolatin B (**1**) [8], karanjin (**2**) [8], 2″,2″-dimethylpyrano-[5″,6″:7,8]-flavone (**3**) [8], and pongamol (**4**) [4,18]. The purity of the isolated compounds (>95%) was determined based on HPLC-PDA analysis. These isolates were used as standards for quantitative analysis.

*Lanceolatin B (**[2″,3″:7,8]-furanoflavone)* (**1**): white needle crystal; C_17_H_10_O_3_; HR-ESI-MS [M + H]^+^ *m/z* 263.0703 (calcd. for C_17_H_11_O_3_, 263.0708); UV (MeOH) λ_max_: 215, 264, 298 nm; ^1^H NMR (500 MHz, CDCl_3_) δ_H_ 8.21 (d, *J* = 8.8, 1H, H-5), 8.00 (dd, *J* = 6.0, 3.7, 2H, H-2′,6′), 7.81 (d, *J* = 2.2, 1H, H-2″), 7. 61 (d, *J* = 0.6, 1H, H-6), 7.59 (d, *J* = 2.6, 2H, H-3′,5′), 7.43 (d, *J* = 1.0, 1H), 7.25 (dd, *J* = 2.1, 0.6, 1H, H-3″), 7.24 (s, 1H, H-3); ^13^C NMR (125 MHz, CDCl_3_) δ_C_ 178.3, 162.7, 158.4, 150.9, 145.8, 131.8, 131.6, 129.1 × 2, 126.2 × 2, 121.8, 119.4, 117.2, 110.2, 108.1, 104.2 (Appendix A).

*Karanjin* (**2**): white crystal; C_18_H_12_O_4_; HR-ESI-MS [M + H]^+^ *m/z* 293.0808 (calcd. for C_18_H_13_O_4_, 293.0813); UV (MeOH) λ_max_: 217, 259, 304 nm; ^1^H NMR (500 MHz, CDCl_3_) δ_H_ 8.23 (d, *J* = 8.9, 1H, H-5), 8.17 (dd, *J* = 8.3, 1.5, 2H, H-2′,6′), 7,78 (d, *J* = 2.2, 1H, H-2″), 7.59 (d, *J* = 3.2, 1H, H-6), 7,58 (d, *J* = 2.1, 2H, H-3′,5′), 7.57 (d, *J* = 1.3, 1H, H-4′), 7.20 (dd, *J* = 2.2, 0.9), 1H, H-3″), 4.00 (s, 3H, OCH_3_); ^13^C NMR (125 MHz, CDCl_3_) δ_C_ 175.1, 158.2, 154.9, 150.0, 145.1, 141.9, 131.0, 130.7, 128.6 × 2, 128.4 × 2, 121.9, 119.7, 117.0, 110.0, 104.3, 60.2 (Appendix A Appendix A).

*2″,2″-Dimethylpyrano-[5″,6″:7,8]-flavone* (**3**): white needle crystal; C_20_H_16_O_3_; HR-ESI-MS [M + H]^+^ *m/z* 305.1175 (calcd. for C_20_H_17_O_3_, 305.1178); UV (MeOH) λ_max_: 220, 271, 325 nm; ^1^H NMR (500 MHz, CDCl_3_) δ_H_ 8.02 (d, *J =* 8.7, 1H, H-5), 7.93 (d, *J =* 2.1, 2H, H-2′,6′), 7.56 (d, *J =* 1.3, 1H, H-4′), 7.55 (d, *J =* 1.9, 2H, H-3′,5′), 6.97 (dd, *J =* 10.0, 0.7, 1H, H-4′′), 6.89 (dd, *J =* 8.7, 0.7, 1H, H-6), 6.78 (s, 1H, H-3), 5.79 (d, *J =* 10.0, 1H, H-3″), 1.54 (s, 6H, CH_3_); ^13^C NMR (125 MHz, CDCl_3_) δ_C_ 177.9, 162.6, 157.6, 152.4, 132.1, 131.4, 130.5, 129.1 × 2, 126.1 × 2, 126.1, 117.8, 115.2, 115.2, 109.5, 107.5, 77.7, 28.2, 28.2 (Appendix A).

*Pongamol* (**4**): white crystal; C_18_H_14_O_4_; HR-ESI-MS [M + H]^+^ *m/z* 295.0967 (calcd. for C_18_H_13_O_4_, 295.0970); UV (MeOH) λ_max_: 241, 348 nm; ^1^H NMR (500 MHz, CDCl_3_) δ_H_ 8.01 (dd, *J* = 8.5, 1.5, 2H, H-2,6), 7.90 (d, *J* = 8.7, 1H, H-6′), 7.65 (d, *J* = 2.3, 1H, H-2″), 7.56 (d, *J* = 7.4, 1H, H-4), 7.52 (d, *J* = 7.7, 2H, H-3,5), 7.34 (dd, *J* = 8.8, 0.8, 1H, H-5′), 7.19 (s, 1H, H-α), 7.03 (dd, *J* = 2.2, 0.8, 1H, H-3″), 4.17 (s, 3H, OCH3); ^13^C NMR (125 MHz, CDCl_3_) δ_C_ 186.1, 184.3, 158,7, 153.8, 144.9, 135.7, 132.2, 128.6 × 2, 127.1 × 2, 126.5, 122.3, 119.7, 107.1 105.3, 97.9, 61.2 (Appendix A).

### 2.2. Simultaneous Quantification of Four Major Compounds

#### 2.2.1. Optimized HPLC Condition

Based on the ultraviolet spectra of compounds **1**–**4**, the wavelength for quantitative determination at 254 nm was chosen to obtain the baseline separation of the markers. A gradient elution using water-acetonitrile as the mobile phase was conducted to successfully separate compounds **1**–**4** in the extract from *M. pulchra* radix within 30 min (Figure 2). The peak purity of these isolates was more than 99.98%.

#### 2.2.2. HPLC-PDA Method Validation

All standards proved linearity with R^2^ > 0.995 at the test concentration range. This result showed good linearity of the developed quantitative method. The limit of detection (LOD) and limit of quantification (LOQ) values were calculated at 254 nm. The LOD for the four analytes ranged from 0.002 to 0.012 µg/mL, while their LOQ was observed in the range 0.020–0.040 µg/mL (Table 1). The RSD value was less than 4% for the within-day and day-to-day precision, and the recovery was observed in the range of 91.76–102.27% and 94.94–104.44%, respectively (Table 2). Six batches of samples were analyzed, and the repeatability was measured with an RSD range of 0.71–1.75%. The samples were injected into the HPLC system at 0, 24, and 48 h to examine the sample solution’s stability. The results showed that the RSDs were in the range of 1.10–3.13% (Table 3). These obtained results showed that the established quantitative method was sensitive, precise, and accurate to determine the four flavonoids in the *M. pulchra* radix simultaneously.

#### 2.2.3. Determination of Major Components in *M. pulchra* Radix

The developed HPLC-PDA method was validated and used to quantify the major components in *M. pulchra* radix at different ages (one to four years old). The peak identification of these components was characterized by comparison with the retention time and UV spectra of the standards. The four (**1**–**4**) analytes’ retention times were 12.4, 14.0, 19.1, and 24.0 min, respectively. All resolution factors of the analyte peaks were greater than 1.5. The contents of the major compounds were determined from the corresponding calibration curve (Table 4). The results proved a remarkable difference in the analytes’ contents between the one-year-old sample and the samples that were more than two years old. The amount of all analytes in *M. pulchra* radix significantly increased when the plant had grown for more than two years. After two years, their contents showed nearly no change (Figure 3).

This is the first report on simultaneous quantification of four major flavonoids (lanceolatin B, karanjin, 2″,2″-dimethylpyrano-[5″,6″:7,8]-flavone, and pongamol) in *M. pulchra* radix. Previously, Fan et al. [19] determined the karanjin, about 0.50–0.65 mg/g in *M. pulchra* radix, by employing coupling of ultra-performance liquid chromatography (UPLC) with triple-quadrupole mass spectrometry (QqQ-MS). However, the study lacks the investigation on lanceolatin B, 2″,2″-dimethylpyrano-[5″,6″:7,8]-flavone, and pongamol. The other report investigated the application of HPLC-DAD-ESI-IT-TOF-MS^n^ for characterization of twelve flavonoids in *M. pulchra* radix, including lanceolatin B, karanjin, 2″,2″-dimethylpyrano-[5″,6″:7,8]-flavone, and pongamol [20], but it did not present the contents of the mentioned constituents. In our study, a simple and effective method was used to isolate and purify karanjin, lanceolatin B, pongamol, and 2″,2″-dimethylpyrano-[5″,6″:7,8]-flavone from *Millettia pulchra* radix. The HPLC-PDA (Photo Diode-array Detector) method was developed and validated for qualitative and quantitative analysis of these isolates in *M. pulchra* radix at different ages.

### 2.3. Optimization of Extraction

#### 2.3.1. RSM Modelling

Extraction time, solvent-to-material ratio and ethanol concentration were found to be vital parameters affecting the extraction efficiency of the major components from *M. pulchra* radix. Table 5 provides the coding of the experimental parameters and related levels for optimization of the extraction. Twenty experiments were conducted and the results of the yield, contents of the major components are shown in Table 6. The designed models for the five responses were all found to be significant (*p* < 0.0001), thus indicating the relationship between the independent and dependent variables (Appendix A). The values of the regression coefficients were calculated and the fitted-in equations to predict the yield of major compounds’ contents from *M. pulchra* radix were as follows:(1)$$Y_{1}\left(\%\right)=-26.0462+26.42969X_{1}-4.65418X_{2}+1.13137X_{3}+0.7061743X_{1}X_{2}-0.10043X_{1}X_{3}-0.03153X_{3}-4.83888X_{1}^{2}+0.336508X_{2}^{2}-0.00621X_{3}^{2}$$
(2)$$Y_{2}\left(µg/g\right)=28.39333-3.04598X_{1}-0.8243X_{2}-0.72135X_{3}+0.072135X_{1}X_{2}+0.039593X_{1}X_{3}-0.008X_{2}X_{3}+0.150662X_{1}^{2}+0.070693X_{2}^{2}+0.007127X_{3}^{2}$$
(3)$$Y_{3}\left(µg/g\right)=25.38287+11.99595X_{1}-9.31822X_{2}+0.015351X_{3}+0.557899X_{1}X_{2}+0.063076X_{1}X_{3}+0.059673X_{3}-4.57486X_{1}^{2}+0.323837X_{2}^{2}+0.001851X_{3}^{2}$$
(4)$$Y_{4}\left(µg/g\right)=15.84437+2.65877X_{1}-1.0527X_{2}-0.46451X_{3}+0.021711X_{1}X_{2}+0.016496X_{1}X_{3}+0.001888X_{2}X_{3}-0.79727X_{1}^{2}+0.051159X_{2}^{2}+0.003807X_{3}^{2}$$
(5)$$Y_{5}\left(µg/g\right)=16.24879-2.20293X_{1}-0.51456X_{2}-0.37895X_{3}+0.124393X_{1}X_{2}+0.021957X_{1}X_{3}+0.005426X_{2}X_{3}-0.0286X_{1}^{2}+0.000208X_{2}^{2}+0.002433X_{3}^{2}$$

As shown in Appendix A, the models (yield of extraction, contents of major compounds) were significant, as is evident from the F-values with a low probability value (*p* < 0.0001). The high coefficient of determination (R^2^) (0.9679–0.9917) was provided by ANOVA of the quadratic regression models, indicating that the models could explain 96.79–99.17% of the variation in data. Moreover, the lack of fit was significant, suggesting that the regression models were adequate for describing the observed data variations. In addition, the model had an Adequate Precision (AP) greater than 4, which is desirable to indicate adequate model discrimination. AP is the ratio of the signal-to-noise ratio [15]. For the proposed models, the AP values ranged from 21.51 to 36.57, indicating very good signal-to-noise ratios. Figure 4 shows that the second-order polynomial regression models were in significant agreement with the experimental data. The results proposed that the designed models in this study could identify the conducted conditions for the selective extraction of *M. pulchra* radix.

#### 2.3.2. The Influence of Process Variables on The Extraction Yield

The obtained results (Appendix A) revealed that the quadratic effects of the extraction time (X_1_), solvent-to-material ratio (X_2_), and ethanol concentration (X_3_) were remarkable with statistical impact of process variables (*p* < 0.05) on the yield (Y_1_) extracted from *M. pulchra* radix. The results suggested that the extraction process under the condition of X_1_ = 2, X_2_ = 12, and X_3_ = 70% would obtain the maximum yield value (Y_1_ = 24.40%). Figure 5 and Figure 6 showed that the relationship between the independent variables’ interaction effect and the yield value was significantly quadratic in the investigated ranges. The yield increased with an increase in the solvent-to-material ratio and a decrease in the ethanol concentration. *M. pulchra* radix contains many polysaccharides [21]. Thus, at lower ethanol concentrations, polysaccharides can easily migrate to plant tissue, leading to enhanced yield due to the reduced ethanol concentration. The increased solvent diffusivity into plant cells and the enhanced desorption of components can be explained by the increased yield due to the high solvent-to-material ratio and extraction time [22].

#### 2.3.3. The influence of Process Variables on The Major Components’ Contents

The obtained results indicated that the quadratic effects of all independent variables were not similar on all responses. The quadratic effects of the extraction time, the ratio of solvent, material, ethanol extraction were significant (*p* < 0.001) on the contents of compounds **2** and **4,** whereas only extraction time as well as ethanol concentration influenced significantly (*p* < 0.05) on the contents of **1** and **3**. According to the analyzed results, the optimum values of the independent variables for the maximum contents of the major compounds extracted from *M. pulchra* radix were X_1_ = 1, X_2_ = 7, X_3_ = 95% for **1**, X_1_ = 3, X_2_ = 12, X_3_ = 95% for **2**, and X_1_ = 1, X_2_ = 12, X_3_ = 95% for both **3** and **4**.

Three-dimensional response surface plots and contour scatter plots (Figure 5 and Figure 6) were provided according to Equations (1)–(5) to predict the interaction between the independent variables and the responses. Increasing the concentration of ethanol and the extraction time resulted in an increase in the major components’ contents. The major compounds from *M. pulchra* radix belonged to flavonoid derivatives. Therefore, when increasing the ethanol ratio in water and duplicating the process, the extraction process gave rise to the extraction of flavonoids [15]. As for the solvent-to-material ratio, all dependent parameters went up by increasing the solvent quantity. However, the solvent quantity and extraction time interaction slightly influenced the extraction of major compounds.

#### 2.3.4. ANN Modelling

The extraction experiments conducted to develop the RSM model were also used to design the ANN model: 70% (14 points) for training network, 15% (3 points) for validation, and the remainder 15% (3 points) for the testing network. The network training was conducted by using the Levenberg–Marquardt algorithm. The number of neurons in the hidden layer was selected after achieving a minimum error of the predictive models and obtained an optimal ANN model with minimal dimension and error in training and testing network. In this study, the best ANN model with a 3-10-5 topology was established after verifying various learning algorithms (Figure 7). Figure 8 shows the significant fitting in the neural network between experimental and predictive data for training, testing, and validation. Furthermore, the coefficient of determination and root mean square error were observed in the ranges of 0.9771–0.9951 and 0.22–0.74, respectively. It means that the ANN modelling fitted well with experimental data.

#### 2.3.5. Verification and Comparison Between RSM and ANN Models

The performance of RSM and ANN models was compared concerning their predictive performance and estimation capabilities. The coefficient of determination (R^2^) and root mean square error (RMSE) of both RSM and ANN models were provided in Table 7. Both RSM and ANN modellings proved an excellent quality of predictions. However, the ANN approach showed higher values of R^2^ and lower values of RMSE. These results suggested that ANN modelling can effectively predict the responses with higher capabilities when compared to the RSM model. The ANN approach’s improved accuracy may be associated with its universal ability to approximate nonlinear systems. In comparison, the RSM model is only based on the second-order polynomial regression [18]. Several studies that compared the RSM and ANN modellings for efficient extraction of natural products reported that ANN architecture was superior to the RSM in predicting process recovery [17,18,19]. Thus, the ANN model can predict the experimental data for all responses (Y_1–5_) with reliable precision and validity.

As shown in Table 8, the optimal condition of the extraction process for *M. pulchra* radix obtained by the ANN approach was almost identical to RSM. The optimal condition corresponded to the extraction time (twice), solvent/material ratio (9.5), and ethanol concentration (72.5%). Conducted experiments were based on the optimal condition. The ANN model suggested the optimum obtained yield of 20.05% for ethanol extract and the major contents of 3.99, 47.01, 4.25, and 1.10 (µg/g), corresponding to compounds **1**–**4**, respectively. The experimental value of ethanol extract and **1**–**4** were determined as 20.05 ± 0.07%, 4.05 ± 0.05 (µg/g), 46.99 ± 0.22 (µg/g), 4.22 ± 0.19 (µg/g), and 1.05 ± 0.05 (µg/g), respectively. These experimental values were similar to the optimal values predicted by the RSM- and ANN-designed models. However, the practical value was more identical to ANN predictive value.

According to our knowledge, this is the first report on the comparison of RSM and ANN for the optimization of major components from *M. pulchra*. The extraction condition of these compounds was primarily optimized by statistically comparing estimated capabilities and efficiencies of RSM and ANN models. The obtained results could be helpful in the pharmaceutical development of this natural resource.

## 3. Materials and Methods

### 3.1. Chemicals and Reagents

The chemical solvents used for extraction and isolation were ethanol, ethyl acetate, dichloromethane, and methanol, which were purchased from Duksan pure chemicals Co., Ltd. (Seoul, Korea). The acetonitrile and methanol used for the HPLC analysis were purchased from Honeywell Inc. (Charlotte, North Carolina, USA). The chromatography columns were conducted by using 200–300 mesh silica gel (Merck, Darmstadt, Germany). NMR spectra were recorded on Bruker Avace spectrometers at 500 MHz for ^1^H and at 125 MHz for ^13^C, using chloroform-*d* as a solvent and tetramethyl silane as a reference. HR-ESI-MS data were obtained on a Finnigan MAT-95XL mass spectrometer combined with a UPLC system (Thermo Fisher Scientific, Bermen, Germany).

The reference compounds, lanceolatin B ([2″,3″:7,8]-furanoflavone) (**1**), karanjin (**2**), 2″,2″-dimethylpyrano-[5″,6″:7,8]-flavone (**3**), and pongamol (**4**), were isolated from *Millettia pulchra* radix. Their purities were determined to be more than 95% by HPLC analysis.

### 3.2. Plant Material

The radix of *Millettia pulchra* Kurz (Fabaceae) was collected in September 2019 in An Giang Province, Vietnam (10.489791° N, 104.978178° E), and authenticated by Instructor T.H. Vo. Samples were harvested at different ages (one, two, three, and four years old). The plant’s voucher specimens (VMP201909A-D) were deposited at the Herbarium of the National Research Institute of Chinese Medicine, Ministry of Health and Welfare, Taiwan (R.O.C.).

The materials were cut into slices approximately 2–5 mm thick, then dried at 50 °C until the loss from drying was not more than 15%. The materials were then packaged into plastic bags under a vacuum and stored at a temperature ≤30 °C and a humidity ≤75%.

### 3.3. Extraction and Isolation

The dried three-year-old radix (4.0 kg) of *M. pulchra* was extracted with 95% ethanol at 50 °C for four hours two times. The crude extract (730 g) was yielded by combining and concentrating the filtrate under a vacuum at 40 °C. The crude extract was then suspended in distilled water and partitioned with ethyl acetate (EtOAc) to obtain 200 g of EtOAc residue.

Using the mixture of dichloromethane-methanol (8:2, *v*/*v*), the EtOAc residue was dissolved and then sonicated and filtered. The filtrate was collected and stabilized at 15 °C to obtain 4.0 g of compound **2** in a crystal form. The EtOAc residue solution was evaporated to obtain residue A. By operating the silica gel open column chromatography, residue A was separated into five fractions (A1–5). After dissolving all fractions in dichloromethane, the solution was kept at 15 °C to yield the crystals of compound **1** (1.5 g) from fraction A1, compound **3** (700 mg) from fraction A3, and compound **4** (650 mg) from fraction A4.

The purity of the compounds was analyzed using HPLC-PDA, and their structures were elucidated based on MS and 1D- and 2D-NMR data.

### 3.4. Simultaneous Quantification of Four Major Compounds in M. pulchra Radix

#### 3.4.1. Sample Preparation

The dried raw samples were ground into a powder and filtered through a No. 20 sieve. Exactly 1.0 g of the powdered material was extracted with 25 mL of acetonitrile by sonicating for 60 min at room temperature. The extract was filtered under a vacuum using Whatman No. 1 paper and put into a 25 mL volumetric flask containing acetonitrile. The filtrate was filtered additionally through a 0.22 µm membrane for analysis.

Next, 10 µL of the filtrate were injected into the HPLC-PDA system and immediately analyzed to quantity the marker compounds. The contents of the markers were calculated based on the corresponding calibration curve.

#### 3.4.2. Reference Solutions

The isolated compounds were accurately weighed and dissolved in acetonitrile to obtain stock solutions at 1.0 mg/mL. The stock solutions were then diluted with acetonitrile to get different concentrations in the range of 5–120 µg/mL for compounds **1** and **2**, 10–120 µg/mL for compound **3**, and 5–80 µg/mL for compound **4**. All standard solutions were stored as reference solutions at a temperature under 10 °C and were filtered through a 0.22 µm membrane before being injected into the HPLC system.

#### 3.4.3. HPLC Apparatus and Conditions

The major components of *M. pulchra* radix were determined using the HPLC-PDA system, which consisted of a Shimadzu HPLC apparatus (Shimadzu, Kyoto, Japan) combined with a quaternary pump, a photodiode array detector, and a Cosmosil C_18_ column (i.d. 4.6 mm × 150 mm, 5 µm, Nacalai Tesque Company, Kyoto, Japan). The mobile phase of the water (A) and acetonitrile (B) was set at the following gradient conditions: 0.0 min, 40% B; 20.0–25.0 min, isocratic elution with 80% B; and 35.0–40.0 min, 100% B. The column was re-equilibrated for five min before the next injection. The flow rate was 1.0 mL/minute, and the injected volume was 10 µL. The column temperature was set at 35 °C For detection, the wavelength was set at 254 nm, and the UV spectra were recorded between 200 nm and 400 nm. The major components were determined by comparing their retention time and UV spectra with those of the standards.

#### 3.4.4. Method Validation

According to ICH guidelines [23], the HPLC-PDA method was validated, including the linear range, the limit of detection (LOD), and the limit of quantification (LOQ), as well as the precision, repeatability, stability, and recovery.

For the calibration curve, six different concentrations of each isolated compound were prepared. The calibration curves were plotted by evaluating six dilution concentrations of each analyte as the abscissa and the corresponding peak areas as the coordinate. The LOD and LOQ were calculated at a signal-to-noise ratio (S/N) of 3 and 10, respectively. These values illustrated the lowest concentration of analytes for detection and quantification.

Intra- and inter-day validation were used to measure the precision of the quantitative method. The intra-day variation was calculated by analyzing three exact mixed standard solutions at high, medium, and low concentrations of the major compounds in *M. pulchra* radix in one batch. The inter-day validation was determined through analysis of the three exact mixed standard solutions on three consecutive days.

The three concentration levels of the standards (low, medium, and high) were added to the powdered material. Samples were then prepared according to the above sample preparation process and HPLC was used to analyze the filtrates to measure the recoveries, which were determined by the following equation:(6)$$\mathrm{Recovery}\;(\%)=\frac{\mathrm{total}\;\mathrm{amount}\;\mathrm{detected}-\mathrm{amount}\;\mathrm{orginal}}{\mathrm{spiked}\;\mathrm{amount}\;}\times 100$$

#### 3.4.5. Simultaneous Quantification of Major Compounds in *M. pulchra* Radix

Four sample solutions of *M. pulchra* radix at 1, 2, 3, and 4 years old were injected into the HPLC system. The contents of the four markers were calculated from the corresponding calibration curve.

### 3.5. Optimization of Extraction

#### 3.5.1. Sample Extraction

*M. pulchra* radix at two years old were collected for sample optimization. A powder sample (5.0 g) was mixed with solvent in a 100 mL Erlenmeyer flask and extracted at 50 °C for 60 min. The extract was filtered through Whatman No. 1 paper and then combined with the filtrate and evaporated under a vacuum. The condensed filtrated sample was weighed and dissolved in 50 mL of acetonitrile in a 50 mL marked volumetric flask. After that, the filtered sample was put through a 0.22 µm membrane for quantitative analysis.

The HPLC analysis of the products extracted from *M. pulchra* radix followed the above condition of simultaneous determination of the major compounds.

The concentrations of the major compounds in samples were determined based on the corresponding calibration curve of each compound.

#### 3.5.2. Experimental Design

The effects of different solvents, extraction times, and solvent/material ratios were investigated based on the yield of the extractive substances. The central composite design (CCD) was created using Design-Expert 6.0.6 software to explore the effects of three independent variables (extraction time, solvent/material ratio, and ethanol concentration) on the dependent variables (the yield of the raw extracts and that of the major compounds).

Twenty experiments were carried out, and each experiment was carried out three times to determine the yield of the crude extract and the major compounds’ contents.

The yield of extraction was calculated as:(7)$$Y_{1}\left(\%\right)=\frac{\mathrm{weight}\;\mathrm{of}\;\mathrm{residues}\;\mathrm{after}\;\mathrm{concentrated}\;\left(g\right)}{\mathrm{weight}\;\mathrm{of}\;\mathrm{material}}\times 100$$

The contents of the major components in the yield were calculated as:(8)$$X_{i}\left(µg/g\right)=\frac{\mathrm{weight}\;\mathrm{of}\;\mathrm{major}\;\mathrm{compound}\;\mathrm{in}\;\mathrm{yield}\;\left(µg\right)}{\mathrm{weight}\;\mathrm{of}\;\mathrm{yield}\;\left(Y_{1}\right)\;\left(g\right)}.\;$$

The quadratic effects of the three variables under research and their interactions on the extraction yield and contents of the major components of *M. pulchra* radix were determined by Design-Expert software. The ANOVA test was then used to analyze the results and specify their significance. A polynomial equation fitted the obtained results to correlate the response to the independent variables. The general equation to predict the optimal condition was as follows [24]:(9)$$Y=\alpha_{k0}+\overset{3}{\underset{i=1}{\sum }}\alpha_{\mathrm{ki}}x_{i}+\overset{3}{\underset{i=1}{\sum }}\alpha_{\mathrm{kii}}x_{i}^{2}+\overset{3}{\underset{i<j=2}{\sum }}\alpha_{\mathrm{kij}}x_{i}x_{j}$$
in which, Y is the predicted response; α_k0_, α_ki_, α_kii_, and α_kij_ are represent regression coefficients; *x_i_* and x_j_ are the coded independent factors.

The goodness of fit of the constructed polynomial models was calculated based on the regression coefficient (R^2^), adjusted-R^2^ (R^2^_adj_), the prediction error sum of squares (PRESS), and the adequate precision (AP) [24].

All results were provided as a mean of three values with the standard deviation.

#### 3.5.3. Artificial Neural Network (ANN) Modelling

In this study, a commercial ANN software, MATLAB—R2021a (MathWorks Inc., Natick, MA, USA), was used to control the nonlinear relationship between independent variables and responses. The network architecture comprised one input layer (extraction time, solvent-to-material ratio, and ethanol concentration), one hidden layer, and one output player (yield of extraction and contents of major components). CCD decided the number of neurons. The entire data set of 20 extraction experiments was divided into three sets: 70% of CCD design for the training dataset and 15% of CCD design for each validation and testing dataset. A multilayer feed-forward neural network trained with an error back-propagation algorithm was conducted to model the target responses. Multilayer feed-forward neural network of various topologies was tested by determining the number of neurons in the hidden layer to obtain an optimal ANN topology, with the highest coefficient of determination (R^2^) and the lowest mean squared error (MSE) [15,16].

#### 3.5.4. Verification Experiments, Comparison Between RSM and ANN Models

A new set of experimental condition combinations was carried out to verify the accuracy of the statistical results. The experimental and predicted data were finally analyzed and compared to determine the polynomial model’s validity.

The statistical parameters, including coefficient of determination (R^2^) and root mean squared error (RMSE), were determined to compare the predictive performance and estimation capabilities of RSM and ANN models [16].

## 4. Conclusions

The current study is the first report on the quantification and optimization for major components’ extraction from *Millettia pulchra* radix. The results revealed that *M. pulchra* is a potential natural resource for the isolation lanceolatin B, karanjin, 2″,2″-dimethylpyrano-[5″,6″:7,8]-flavone and pongamol by an efficient method. These isolates could be used as marker compounds for quality control of its radix. A rapid, accurate, convenient, reliable HPLC–PDA detection method was established and successfully applied for the quantitative analysis of four marker components in *M. pulchra* radix. Moreover, Response Surface Methodology (RSM), Artificial Neural Network (ANN) were compared their predictive performance and estimation capabilities. The results revealed that the ANN model is more efficient and accurate to optimize extraction conditions leading to maximize the contents of the major compounds [lanceolatin B (**1**), karanjin (**2**), 2″,2″-dimethylpyrano-[5″,6″:7,8]-flavone (**3**), and pongamol (**4**)]. Hence, the current strategy suggests its potential in pharmaceutical developments of the titled plant.

## Figures and Tables

**Figure 1 molecules-26-03641-f001:**
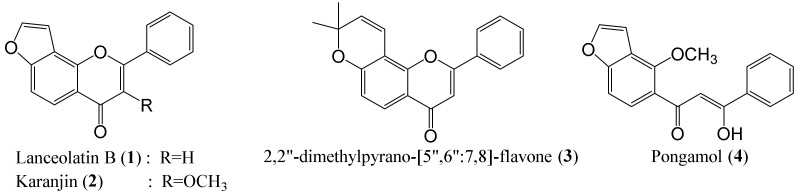
Major isolates from *M. pulchra* Kurz radix.

**Figure 2 molecules-26-03641-f002:**
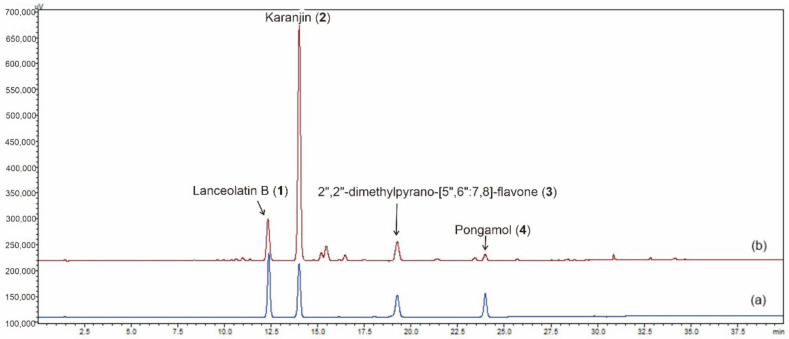
The HPLC-PDA diagram of standard mixture (**a**) and *M. pulchra* radix (**b**).

**Figure 3 molecules-26-03641-f003:**
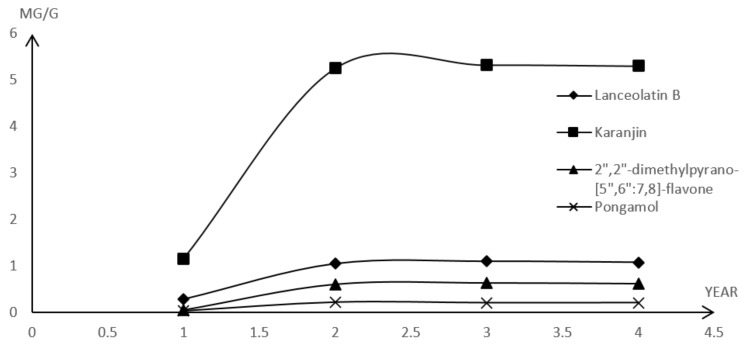
The contents of major compounds in *M. pulchra* radix at different ages.

**Figure 4 molecules-26-03641-f004:**
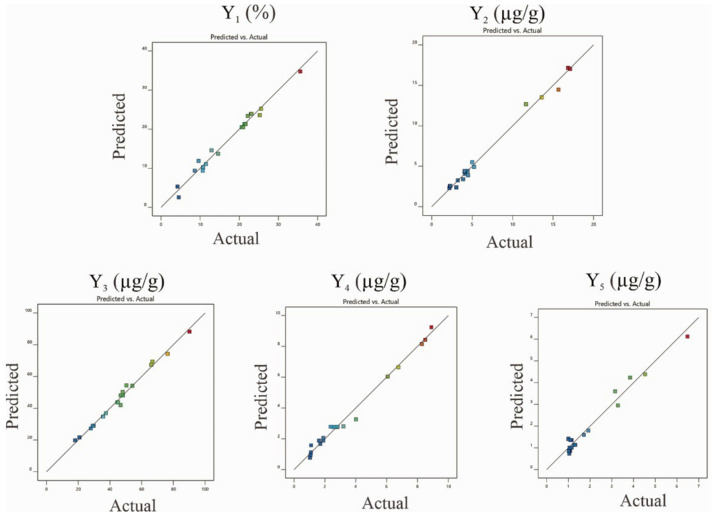
Comparison of experiment data with the predicted value obtained by the RSM models. (Y_1_: The yield of extraction (%); Y_2–5:_ The contents of major components **1**–**4**, respectively).

**Figure 5 molecules-26-03641-f005:**
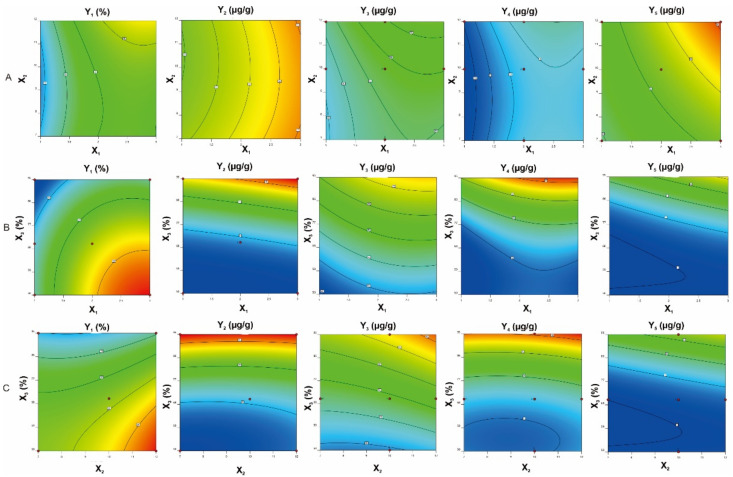
Contour plots are illustrating the relationships between independent variables and responses. (**A**)—Extraction time (X_1_) compared to solvent-to-material ratio (X_2_); (**B**)—Extraction time (X_1_) compared to ethanol concentration (X_3_); (**C**)—Solvent-to-material ratio (X_2_) compared to ethanol concentration (X_3_); Y_1_: The yield of extraction; Y_2–5_: The contents of major components **1**–**4**, respectively).

**Figure 6 molecules-26-03641-f006:**
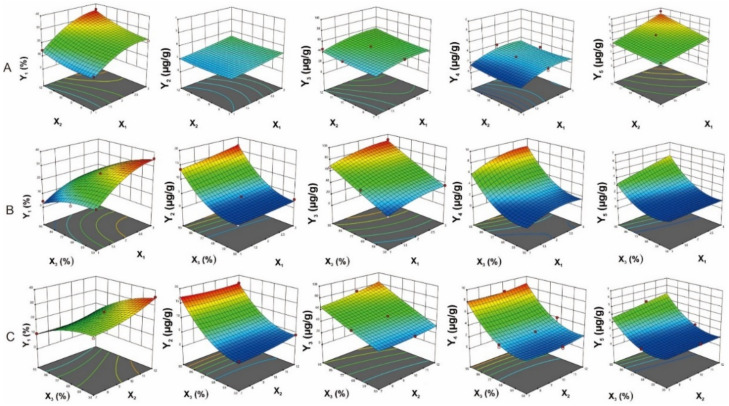
Three-dimensional response surface plots are illustrating the relationships between independent variables and responses. (**A**)—Extraction time (X_1_) compared to solvent-to-material ratio (X_2_); (**B**)—Extraction time (X_1_) compared to ethanol concentration (X_3_); (**C**)—Solvent-to-material ratio (X_2_) compared to ethanol concentration (X_3_); Y_1_: The yield of extraction; Y_2–5_: The contents of major components **1**–**4**, respectively).

**Figure 7 molecules-26-03641-f007:**
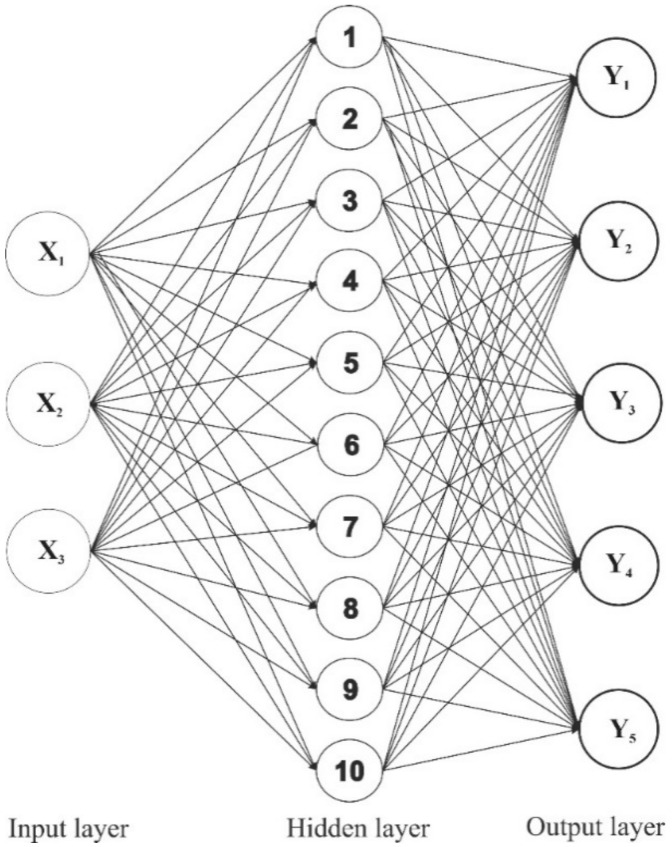
The optimal architecture of the ANN model. (X_1_: Extraction time; X_2_: Solvent-to-material ratio; X_3_: Ethanol concentration; Y_1_: The yield of extraction (%); Y_2–5_: The contents of major components **1**–**4**, respectively).

**Figure 8 molecules-26-03641-f008:**
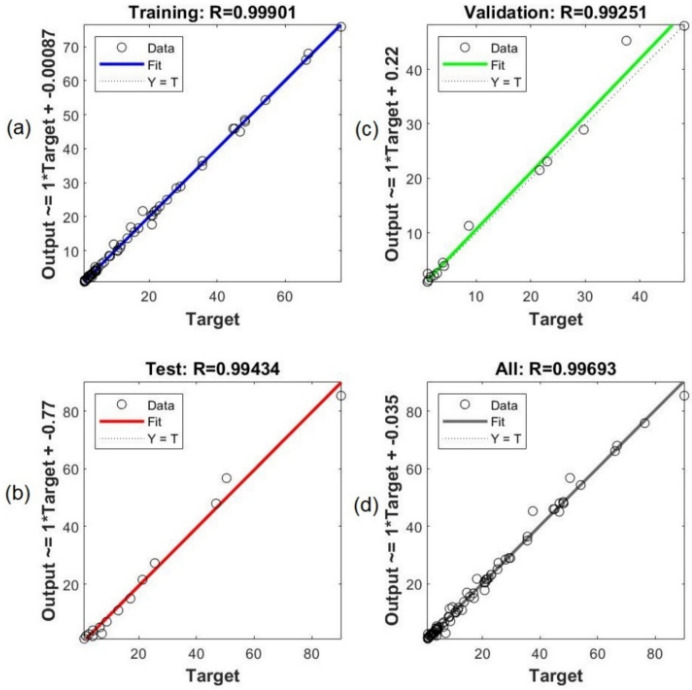
Scatter plot between experimental and predictive data by ANN modeling for (**a**) training, (**b**) testing, (**c**) validation, and (**d**) overall data fitting.

**Table 1 molecules-26-03641-t001:** Calibration parameters of HPLC-PDA analysis of four markers.

Markers	Retention Time (min)	Regression Equation ^a^	R^2^	Standard Curve (µg/mL)	LOD ^b^ (µg/mL)	LOQ ^c^ (µg/mL)
Lanceolatin B	12.4	ŷ = 46992523.93x − 161288.07	0.9976	5–120	0.006	0.020
Karanjin	14.0	ŷ = 40497797.83x	0.9984	5–120	0.006	0.020
2,2-Dimethylpyrano-[5,6:7,8]-flavone	19.1	ŷ = 41233664.76x	0.9976	10–130	0.012	0.040
Pongamol	24.0	ŷ = 27135799.19x	0.9998	5–80	0.002	0.006

^a^ in the regression equation ŷ = ax + b, x and y indicate the concentration (µg/mL) and peak area (mAU.s), respectively. R^2^ is the correlation coefficient of the equation. ^b^ LOD = 3 × signal-to-noise ratio. ^c^ LOQ = 10 × signal-to-noise ratio.

**Table 2 molecules-26-03641-t002:** Intra- and inter-day precision and recovery.

Markers	Theoretical Conc. (µg/mL)	Found Conc. (µg/mL)		RSD (%)		Recovery (%)	
		Intra-Day ^a^	Inter-Day ^b^	Intra-Day ^a^	Inter-Day ^b^	Intra-Day ^a^	Inter-Day ^b^
Lanceolatin B	40	36.70	37.98	0.55	3.08	91.76	94.94
	60	58.67	58.74	1.43	2.25	97.78	97.89
	80	77.87	79.67	0.39	2.07	97.33	99.59
Karanjin	40	37.06	38.76	1.92	3.94	92.65	96.90
	60	61.36	61.11	1.22	0.57	102.27	101.84
	80	78.31	77.38	1.15	1.13	97.88	96.73
2″,2″-Dimethylpyrano-[5″,6″:7,8]-flavone	40	42.49	41.78	1.04	1.82	106.23	104.44
	60	60.28	60.16	0.75	0.53	100.47	100.27
	80	81.73	82.25	0.69	0.73	102.16	102.82
Pongamol	20	19.82	19.86	3.77	1.59	99.12	99.28
	40	40.46	40.71	1.37	0.83	101.16	101.77
	60	60.50	59.99	0.59	1.04	100.83	99.98

^a^ Intra-day at three times in one day. ^b^ Inter-day on three different days.

**Table 3 molecules-26-03641-t003:** The relative standard deviations of standards’ precision and repeatability and stability of sample solution.

Markers	Precision RSD (%)		Repeatability RSD ^c^ (%)	Stability RSD (%) ^b,^*
	Intra-Day ^a^	Inter-Day ^b^		
Lanceolatin B	0.07	1.13	1.04	1.53
Karanjin	0.37	0.79	0.71	1.10
2,2-Dimethylpyrano-[5,6:7,8]-flavone	0.52	1.59	1.75	3.13
Pongamol	0.13	0.56	0.96	1.54

^a^ Intra-day at three times in one day. ^b^ Inter-day on three different days. ^c^ Six batches per day. * Stored at under 10 °C.

**Table 4 molecules-26-03641-t004:** Contents of four major components in *M. pulchra* radix at different ages.

*M. pulchra* Radix	Age (Year)	Contents [(µg/g), n = 3 (RSD)]			
		Lanceolatin B	Karanjin	2″,2″-Dimethylpyrano-[5″,6″:7,8]-flavone	Pongamol
Sample A	1	0.2946 (0.14)	1.1521 (1.17)	0.0519 (0.87)	0.0376 (0.35)
Sample B	2	1.0499 (0.44)	5.2534 (0.44)	0.6021 (1.00)	0.2194 (0.24)
Sample C	3	1.1002 (0.18)	5.3234 (0.19)	0.6321 (0.12)	0.2079 (0.43)
Sample D	4	1.0793 (0.32)	5.3017 (0.67)	0.6150 (0.30)	0.2100 (0.20)

**Table 5 molecules-26-03641-t005:** Coding of experimental parameters and related levels.

Independent Variables	Unit	Symbols	Code Values		
			−1	0	+1
Extraction time	-	X_1_	1	2	3
Solvent-to-material ratio	mL/g	X_2_	7	10	12
Concentration of ethanol	(%)	X_3_	50	70	95

**Table 6 molecules-26-03641-t006:** The predictive and experimental data of yield and contents of major compounds extracted from *M. pulchra* radix.

Run	Independent Variables			Yield (%)			Content of Major Components (µg/g)											
	X_1_	X_2_	X_3_	Y_1, exp_^a^	Y_1, pre._^b^	Y_1, pre._^c^	Y_2, exp_. ^a^	Y_2, pre._^b^	Y_2, pre._^c^	Y_3, exp_. ^a^	Y_3, pre._^b^	Y_3, pre._^c^	Y_4, exp_. ^a^	Y_4, pre._^b^	Y_4, pre._^c^	Y_5, exp_. ^a^	Y_5, pre._^b^	Y_5, pre._^c^
1	2	7	70	20.58 ± 1.34	20.48	20.56	4.42 ± 0.18	4.69	4.43	44.67 ± 1.20	43.95	44.66	2.37 ± 0.90	2.40	2.48	1.15 ± 0.03	1.09	1.00
2	2	10	95	11.46 ± 1.18	10.75	11.48	15.66 ± 1.00	13.83	15.70	76.39 ± 1.97	74.91	76.40	8.28 ± 0.43	8.53	8.20	4.53 ± 0.22	4.39	4.59
3	2	12	70	25.56 ± 1.24	25.40	25.56	5.23 ± 0.90	4.98	5.21	50.44 ± 2.00	54.42	50.45	4.01 ± 0.09	3.19	4.01	1.71 ± 0.12	1.43	1.75
4	2	10	50	23.00 ± 2.30	24.30	22.99	2.24 ± 0.72	2.54	2.27	29.69 ± 0.92	28.59	29.69	1.62 ± 0.10	1.85	1.75	1.25 ± 0.05	1.78	1.04
5	1	7	95	4.22 ± 0.82	4.61	4.22	13.59 ± 0.98	13.60	13.59	54.16 ± 1.56	54.47	54.16	6.07 ± 0.10	5.94	6.07	3.28 ± 0.80	2.95	3.28
6	3	12	50	35.58 ± 2.31	34.90	35.58	4.14 ± 0.02	4.07	4.14	35.66 ± 1.00	34.81	35.67	1.90 ± 0.21	2.16	1.88	1.00 ± 0.12	1.32	1.02
7	3	7	95	10.73 ± 1.16	10.86	10.72	16.86 ± 1.00	17.41	16.84	66.08 ± 2.00	66.72	66.09	8.50 ± 0.78	8.38	8.53	3.84 ± 0.38	4.23	3.82
8	1	10	70	8.63 ± 1.11	9.27	8.63	3.88 ± 0.22	3.66	3.88	37.50 ± 0.90	37.01	37.50	1.09 ± 0.05	1.28	1.11	1.04 ± 0.09	1.03	1.02
9	2	10	70	21.60 ± 1.15	21.35	21.34	4.12 ± 0.91	4.22	4.04	48.12 ± 1.02	48.34	46.81	2.82 ± 0.10	2.71	2.57	1.09 ± 0.07	1.33	1.26
10	2	10	70	21.32 ± 2.20	21.35	21.34	4.05 ± 0.12	4.22	4.04	46.79 ± 1.25	48.34	46.81	2.69 ± 0.28	2.71	2.57	1.10 ± 0.10	1.33	1.26
11	2	10	50	23.10 ± 2.30	24.31	22.99	2.3 ± 0.08	2.54	2.27	29.21 ± 1.25	28.59	29.69	1.89 ± 0.12	1.85	1.75	1.32 ± 0.12	1.18	1.04
12	1	7	50	10.68 ± 2.00	10.00	10.68	2.19 ± 0,11	2.21	2.19	18.10 ± 1.90	18.98	18.10	1.03 ± 0.10	0.89	0.98	1.00 ± 0.10	1.19	1.06
13	3	7	50	22.16 ± 1.21	22.94	22.20	3.04 ± 0.42	2.70	3.00	28.00 ± 0.97	27.98	28.12	1.70 ± 0.08	1.75	1.67	1.04 ± 0.05	0.93	1.13
14	1	12	50	14.60 ± 1.29	13.39	14.27	3.22 ± 0.50	3.08	2.61	20.79 ± 1.10	22.22	21.23	1.07 ± 0.05	0.95	2.14	1.08 ± 0.02	0.98	1.91
15	2	10	70	21.54 ± 1.16	21.35	21.34	4.10 ± 0.22	4.22	4.04	48.20 ± 1.00	48.34	46.81	2.550.10	2.71	2.57	1.15 ± 0.14	1.33	1.26
16	3	12	95	12.91 ± 2.19	14.09	12.90	17.10 ± 0.62	17.27	17.09	90.17 ± 1.55	87.86	90.18	8.90 ± 0.28	9.55	8.95	6.48 ± 0.11	6.22	6.44
17	3	10	70	25.23 ± 1.31	23.35	25.24	5.00 ± 0.37	5.20	5.03	48.08 ± 2.00	50.47	48.08	3.20 ± 0.10	2.62	3.17	1.92 ± 0.21	1.60	1.93
18	2	7	70	21.00 ± 1.05	20.48	20.56	4.08 ± 0.21	4.34	4.43	45.12 ± 2.30	43.94	44.66	2.81 ± 0.21	2.81	2.48	1.05 ± 0.17	1.09	1.00
19	1	12	70	9.61 ± 1.18	11.15	12.87	4.49 ± 0.07	4.30	3.44	46.70 ± 1.27	41.70	46.70	1.10 ± 0.03	1.56	1.16	1.01 ± 0.05	1.00	1.07
20	1	12	95	4.52 ± 0.93	3.93	4.52	11.65 ± 0.87	12.71	11.65	66.80 ± 0.96	69.42	66.79	6.76 ± 0.10	6.32	6.76	3.15 ± 0.10	3.50	3.15

X_1_: Extraction time, X_2_: Ratio of solvent/material (mL/g), X_3_: Concentration of ethanol (%); Y_1, exp._, Y_1, pre._: Experimental and predicted yield of extraction (%), Y_2–5, exp._, Y_2–5, pre_: Experimental and predicted content of major components **1**–**4**, respectively. ^a^ Mean ± standard deviation (*n* = 3); ^b^ Predictive value from RSM; ^c^ Predictive value from ANN.

**Table 7 molecules-26-03641-t007:** Comparison between RSM and ANN modellings.

Parameters	RSM					ANN				
	Y_1_	Y_2_	Y_3_	Y_4_	Y_5_	Y_1_	Y_2_	Y_3_	Y_4_	Y_5_
R^2^	0.9875	0.9917	0.9889	0.9767	0.9678	0.9923	0.9951	0.9931	0.9918	0.9771
RMSE	0.87	0.53	1.89	0.33	0.25	0.74	0.28	0.47	0.26	0.22

**Table 8 molecules-26-03641-t008:** The yields of extraction and **1**–**4** predicted by RSM and ANN models with optimal condition ^*^ and compared with experimental data.

Y_1_ (%)			Y_2_ (µg/g)			Y_3_ (µg/g)			Y_4_ (µg/g)			Y_5_ (µg/g)		
RSM	ANN	Exp. ^a^	RSM	ANN	Exp.	RSM	ANN	Exp.	RSM	ANN	Exp.	RSM	ANN	Exp.
20.22	20.05	20.05 ± 0.07	3.96	3.99	4.05 ± 0.05	48.87	46.99	46.99 ± 0.22	4.28	4.25	4.22 ± 0.19	1.22	1.10	1.05 ± 0.05

* The optimal condition: Extraction time (X_1_ = 2), Ratio of solvent/material (X_2_ = 9.5), Concentration of ethanol (X_3_ = 72.5%); Y_1–5_: The contents of major components **1**–**4**, respectively; RSM: Predictive value from RSM; ANN: Predictive value from ANN; Exp.: Experimental value. ^a^ Mean ± standard deviation (*n* = 3).

## Data Availability

The original contributions generated for this study are included in the article; the data presented in this study are available on request from the corresponding author.

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
