# Peer review of "Quantification and Optimization of Ethanolic Extract Containing the Bioactive Flavonoids from Millettia pulchra Radix"

_molecules, 2021, doi:10.3390/molecules26123641_

Round 1

Reviewer 1 Report

This study aimed to isolate the major compounds from M. pulchra radix, develop and validate the HPLC-PDA method to determine their contents, and optimize its extraction. The Response Surface Methodology (RSM) and Artificial Neural Network models (ANN) were compared their predictive performance and optimization of the biochemical extraction process.

The topic of paper is interesting and in the aim of the journal, however the paper requires major revision because needs some additional work and information to be considered for publication.

Specific comments and suggestions for improving the paper are:

 Keywords:  I suggest changing the keyword that are already present in the title (as  Millettia pulchra)

Line 79, please, insert the aims of the study

Lines 80-89, I suggest moving in the discussion

Please, check the chemical structure of pongamol (4)

Figure1, I suggest inserting the name of the bioactive compounds instead of 1,2,3,4.

Table 1, as above

Table 2,3,4 as above

Figure 3, as above

Lines 126, 135, 164, 171, 187, please, insert M. pulchra in italicus.

Reference list

Please check reference 6

Author Response

//

We would like to thank the reviewer for your available comments. We have revised the manuscript step by step, according to your comments. The changes were highlighted in the manuscript and listed as the follows.

REVIEWER #1

  1. Keywords: I suggest changing the keyword that are already present in the title.

Ans.: The keywords: “Millettia pulchra; furanoflavone; pyranoflavone; furanochalcone; Response Surface Methodology, Artificial Neural Network.” has been changed to “Flavonoid; quantification; optimization; Response Surface Methodology, Artificial Neural Network.”

  1. Lines 79: Please, insert the aim of study.

Ans.: Lines 80, the aims of study: “Consequently, our work aims to isolate and quantify the major bioactive flavonoids from M. pulchra radix and optimize the extraction process to obtain the optimal condition for extracting same. Besides that, observing the variation of major components’ content in M. pulchra radix according to the age of plant helps to determine the time for harvesting this medicinal plant.” has been added.

  1. Line 80-89, I suggest moving in the discussion.

Ans.: Line 80-89 “According to our knowledge, this is first report on comparison of RSM and ANN for the optimization of major components from M. pulchra.” Has been moved to Line 303-304 of Results and Discussion section.

  1. Please, check the chemical structure of pongamol (4)

Ans.: Based on the data of Reaxys and SciFinder, together with the literature reference (Chang et al., J. Nat. Prod. 1997, 60, 869-873), the chemical structure of pongamol (4) is correct.  However, the other similar structure (the hydroxyl group is replaced by a carbonyl group) is also named as pongamol in Reaxys. Thus, we would like to name pongamol with its scientific name [(2Z)-3-hydroxy-1-(4–methoxy-1-benzofuran-5-yl)-3-phenylprop 2-en-1-one] for compound 4.

  1. Figures 1-2, tables 1-4, I suggest inserting the name of the bioactive compounds instead of 1, 2, 3, and 4.

Ans.: We have inserted the scientific name of all isolated compounds into the Figures 1-2 and Tables 1-4.

  1. Line 126, 135, 164, 171, 187: please, insert M. pulchra in italics.

Ans.: “M. pulchra” has been revised as “M. pulchra” in line 128, 134, 152, 167, 174, 191, and 209.

  1. Reference list: Please check the reference 6.

Ans.: We have checked the reference 6. Since the paper was belonging to review article, the title name was just presented as “Karanjin”. However, the reference is correct and formatted according to the form of Molecules.

Reviewer 2 Report

- It is important to determine the content of residual solvents that may be present after the extraction, namely, dichloromethane, methanol, in order to be safe in application in pharmaceutical products, in accordance as proposed in paper.

- Given that the HPLC-PDA method is validated, I believe that the robustness must also be validated.
There are articles published on Millettia pulchra that should be referred to in the bibliography, such as: Characterization of Flavonoids in the Ethomedicine Fordiae Cauliflorae Radix and Its Adulterant Millettiae Pulchrae Radix
by HPLC-DAD-ESI-IT-TOF-MSn, 2013, Molecules.
- The discussion should be made considering results obtained previously by other researchers.
- The bibliography must be expanded. because is very short.

Author Response

//

Response Letter to Reviewer 2

We would like to thank the reviewer for your valuable comments. We have revised the manuscript step by step, according to your comments. The changes were highlighted in the manuscript and listed as the follows.

REVIEWER #2

  1. It is important to determine the content of residual solvents that may be present after the extraction, namely, dichloromethane, methanol, in order to be safe in application in pharmaceutical products, in accordance as proposed in paper.

Ans.: In our work, we used ethanol to extract M. pulchra radix to obtain crude extract. Furthermore, for this study, we would like to explore the optimal extraction condition. Thus, we did not mention the determination of the residual solvents’ content. Furthermore, we just used methanol and dichloromethane to isolate the major components which were used as standard compounds for quantification.

  1. Given that HPLC-PDA method is validated, I believe that the robustness must also be validated.

Ans.: In this study, the HPLC-PDA was validated according to ICH guidelines. This guideline did not require to test robustness in the validation method. Following the guide line, limit of detection (LOD), limit of quantification (LOQ), within-day and day-to-day precision, the recovery, repeatability and stability of sample solution were verified. All obtained results showed the relative standard deviations in range of 0.07-3.15%.  These results revealed that the established quantitative method was sensitive, precise, and accurate. Therefore, we did not test the robustness in the study.

  1. There are articles published on Millettia pulchra that should be referred to the bibliography, such as: Characterization of flavonoids in the Ethomedicine Fordiae Cauliflorae Radix and its Adulterant Millettiae Pulchrae Radix by HPLC-DAD-ESI-IT-TOF-MSn, 2013, Molecules.

The discussion should be made considering results obtained previously by other researchers.

Ans.: Besides the above paper, one more reference, “title: Determination of five flavonoids in different parts of Fordia cauliflora by ultra-performance liquid chromatography/triple-quadrupole mass spectrometry and chemical comparison with the root of Millettia pulchra var. laxior. Chem. Cent. J. 2012, 7, 126.” is also added

  1. The bibliography must be expanded. Because it it very short.

Ans.: Two additional references (ref. 13 & 14) about quantification of flavonoids in Millettia pulchra radix are discussed in our manuscript (Line 177-185).

Round 2

Reviewer 1 Report

lines 88-95, I suggest moving in the discussion

Author Response

//

We would like to thank the reviewer so much for your valuable comments.

  1. Lines 88-95, I suggest moving in the discussion

Ans.: Based on your comments, the content of lines 88-95 is divided two parts and removed to the lines 187-191 (In our study, a simple and effective method was used to isolate and purify karanjin, lanceolatin B, pongamol, and 2”,2”-dimethylpyrano-[5”,6”:7,8]-flavone from Millettia pulchra radix. The HPLC-PDA (Photo Diode-array Detector) method was developed and validated for qualitative and quantitative analysis of these isolates in M. pulchra radix at different ages.), and lines 326-329 (the extraction condition of these compounds was primarily optimized by statistically comparing estimated capabilities and efficiencies of RSM and ANN models. The obtained results could be helpful in the pharmaceutical development of this natural resource.), respectively, in the Discussion Section.
